# Effects of *Helicobacter pylori* treatment on the incidences of autoimmune diseases and inflammatory bowel disease in patients with diabetes mellitus

Nai-Wei Sheu[1], Shu-Heng Huang[1], Deng-Chyang Wu[2], John Y. Kao[3☯], Kun-Der Lin[1,4☯]*

1 Department of Internal Medicine, Kaohsiung Municipal Ta-Tung Hospital, Kaohsiung Medical University Hospital, Kaohsiung, Taiwan, 2 Division of Gastroenterology, Department of Internal Medicine, Kaohsiung Medical University Hospital, College of Medicine, Kaohsiung Medical University, Kaohsiung, Taiwan, 3 Division of Gastroenterology, Department of Internal Medicine, Michigan Medicine, University of Michigan, Ann Arbor, Michigan, 4 Division of Endocrinology and Metabolism, Department of Internal Medicine, Kaohsiung Medical University Hospital, College of Medicine, Kaohsiung Medical University, Kaohsiung, Taiwan

☯ These authors contributed equally to this work.
* berg.kmu@gmail.com

**Data Availability Statement:** Data are available at https://figshare.com/s/59dfa39fd1a4877964e8 with the DOI: 10.6084/m9.figshare.19618155.

## Abstract

### Background

*Helicobacter pylori* infection is known to decrease the incidences of autoimmune diseases and inflammatory bowel disease(IBD). Our aim was investigating the effect of *H. pylori* treatment in diabetes mellitus(DM) patients.

### Methods

Adults with newly-diagnosed *H. pylori* infection or peptic ulcer disease(PUD) within the general population and DM population were identified from the National Health Insurance Research Database of Taiwan from 2000–2010. 79,181 patients were assigned to the 3 groups: general population with PUD without *H. pylori* treatment(PUD-HPRx in general population), DM patients with PUD without *H. pylori* treatment(PUD-HPRx in DM), and DM patients with PUD who received *H. pylori* treatment(PUD+HPRx in DM).

### Results

Higher incidences of autoimmune diseases and IBD were observed in the PUD+HPRx in DM group than in the PUD-HPRx in general population and PUD-HPRx in DM groups (autoimmune diseases = 5.14% vs 3.47% and 3.65%; IBD = 5.60% vs 3.17% and 3.25%; P<0.0001). A lower all-cause mortality was noted in the PUD+HPRx in DM group (HR: 0.937, P<0.001) than in the PUD-HPRx in DM group. Trends of a higher incidence of IBD and a lower mortality in younger patients in the PUD+HPRx in DM group compared with the PUD-HPRx in DM group were noted.

**Funding:** This study was funded by a grant from the Ministry of Health and Welfare (MOHW108-TDU-B-212-133006), Kaohsiung Medical University (109CM-KMU- 012), Kaohsiung Medical University Hospital (SA10905), and Kaohsiung Municipal Ta-Tung Hospital (KMTTH-107-014), which had no role in the study design, data analysis, data interpretation, or writing of the manuscript.

**Competing interests:** The authors have declared that no competing interests exist.

**Abbreviations:** CCIS, Charlson's comorbidity index score; HR, Hazard risk; IBD, Inflammatory bowel disease; ICD, International Classification of Disease; NHIRD, National Health Insurance Research Database; NSAIDs, Nonsteroidal anti-inflammatory agents; PUD, Peptic ulcer disease; PUD+HPRx, Patients with peptic ulcer disease and H pylori treatment; PUD-HPRx, Patients with peptic ulcer disease and without H pylori treatment.

## Conclusions

The results revealed that *H. pylori* treatment increased the incidences of autoimmune diseases and IBD and decreased the all-cause mortality in the DM group with PUD. The effect was more significant in younger patients. This finding assists in realizing the influence of *H. pylori* treatment in the DM population.

## Introduction

*Helicobacter pylori* is known to be one of the most common chronic bacterial infection in humans. Based on regional prevalence estimates, there were around 4.4 billion individuals with *H. pylori* infection worldwide in 2015, and over half of the world's population is infected [1]. Evidence suggests that *H. pylori* plays a protective role against several autoimmune diseases. For example, protective effects of *H. pylori* against multiple sclerosis [2], asthma [3, 4], and systemic lupus erythematosus [5] have been reported. A recent study even found that an extract of *H. pylori* effectively inhibited mucus production and various features of inflammation in mice with repeated exposure to house dust mites [6].

An increasing incidence of inflammatory bowel disease (IBD) has been observed in *H. pylori* endemic regions after widespread eradication of *H. pylori* [7]. There exists evidence to suggest a negative association between *H. pylori* infection and inflammatory bowel disease (IBD) [8]. A recent study also indicated that *H. pylori* infection is associated with less fistulizing or stricturing and less-active colitis in Crohn's disease (CD) [9].

Hypotheses regarding the mechanism of protection of *H. pylori* infection have been proposed, including induction of intestinal interleukin (IL)-10 and IL-18-mediated regulatory T cell responses [10] [11]. Several bacterial factors that dampen the immune response of the host were identified, including VacA, CagA, and *H. pylori* lipopolysaccharide [12–14]. Rad et al. found that those infected with *H. pylori* expressed higher levels of Foxp3, which is a regular T cell marker [15]. Depletion of regulatory T cells leads to more severe gastric inflammation and decreases bacterial colonization. Another study revealed that peripheral memory T cells decreased in people infected with *H. pylori*. This reinforced the hypothesis that circulating regulatory T cells in *H. pylori*-infected hosts may exert a protective effect against autoimmune diseases and IBD [16]. Lin et al. [17] investigated the effects of treatment for *H. pylori* infection on the incidences of autoimmune diseases and IBD, and concluded that treatment for *H. pylori* infection is associated with significant increases in the incidences of autoimmune diseases and IBD.

Diabetic mellitus (DM) and *H. pylori* infection are both common in Taiwan. The aim of this study was to determine the effects of *H. pylori* treatment on patients with diabetes.

## Materials and methods

### Data collection

The content and the execution of the current study were approved by the institutional review board of Kaohsiung Medical University Hospital (KMUHIRB-SV(I)-20200018). All methods were carried out in accordance with the guidelines and regulations. Because the analysis was based on de-identified secondary data, individual consent was not required. The consent waiver was approved by the Research Ethics Committee of approved by institutional review board of Kaohsiung Medical University Hospital.

Data from a general population consisting of patients over 18 years of age with newly-diagnosed *H. pylori* infection or peptic ulcer disease (PUD) from 2000–2010, without a prior diagnosis of autoimmune diseases or IBD, were collected from the National Health Insurance Research Database (NHIRD) in Taiwan. Data of patients with DM were collected from the Longitudinal Cohort of Diabetes Patients, which included patients with a diagnosis code indicating DM or those who were prescribed anti-diabetic agents during admission or at an outpatient department. One-hundred and twenty-thousand patients were randomly sampled from newly-diagnosed DM cases every year, and their medical records obtained from the Longitudinal Cohort of Diabetes Patients.

These patients were assigned to the general population or DM group accordingly. The general population patients did not receive *H. pylori* treatment therapy. The DM patients were classified into a group with peptic ulcer disease (PUD) that received treatment for *H. pylori* infection (PUD+HPRx) and a group with PUD without *H. pylori* treatment (PUD–HPRx) (Fig 1).

Subjects were considered to have received *H. pylori* treatment if they were given a course of either triple or quadruple therapy for more than 7 days [17]. *H. pylori* test results for diagnosis or treatment eradication are not available in the NHIRD.

Autoimmune diseases were defined according to International Statistical Classification of Diseases and Related Health Problems 9th Revision (ICD-9) codes as follows: lupus erythematosus (710.0), systemic sclerosis (710.1), rheumatoid arthritis (714.30–714.33), polymyositis (710.4), dermatomyositis (710.3), vasculitis (446.0, 446.2, 446.4, 446.5, 443.1, 446.7, 446.1), pemphigus (694.4), and Sicca syndrome (710.2).

IBD was defined according to ICD-9 codes for Crohn's disease (555.x) and chronic ulcerative colitis (556.x). We added a primary outcome of IBD treated with asacol or azathioprine in order to provide a more strict definition.

## Inclusion criteria

Patients included in this study met the following criteria: newly-diagnosed *H. pylori* infection or peptic ulcer disease (PUD) between 2000 and 2010; ICD-9 code for *H. pylori* (041.86) or

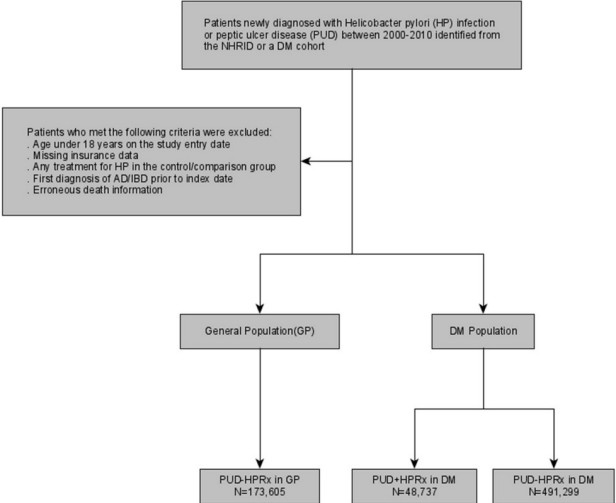

**Fig 1. Flow chart of the study design.** PUD+HPRx, peptic ulcer disease with *Helicobacter pylori* treatment; PUD-HPRx, peptic ulcer disease without *Helicobacter pylori* treatment; DM, diabetes mellitus; AD, autoimmune disease; IBD, inflammatory bowel disease.

PUD (531–533) recorded; received treatment for *H. pylori* infection between 2000 and 2010, with a first treatment date after the study entry date (first date of new diagnosis).

## Exclusion criteria

Patients were excluded from our study if they were aged under 18 years on the date of study entry, had missing insurance enrollment data, received *H. pylori* treatment (in the control group), had a first diagnosis of autoimmune diseases, IBD or a defined event prior to the index date (first date of treatment), or incorrect death information. Index dates were randomly assigned to patients who did not receive *H. pylori* treatment.

The person-years of follow-up were estimated from the index date plus a two-year washout duration to the date of diagnosis of autoimmune diseases or IBD, or an end date of December 31, 2010. We used the two-year washout duration to eliminate the effect of previous *H. pylori* treatment in patients with PUD who received treatment for *H. pylori* infection (PUD+HPRx).

## Statistical analysis

Total 713,641 subjects were included and classified to the 3 study cohorts(PUD-HPRx in general population, PUD-HPRx in DM, and PUD+HPRx in DM). PUD-HPRx in general population group and PUD-HPRx in DM group were matched to PUD+HPRx in DM group using propensity score matching by age, sex, income, comorbidities, and Charlson comorbidity index score. Subjects of the 3 groups after matching were PUD-HPRx in general population group (n = 48,542), PUD-HPRx in DM group(n = 48,737), and PUD+HPRx in DM group (n = 48,737) (Table 1).

The distributions of risk factors in the 3 study cohorts were analyzed by the variance test, chi-square test, or Fisher's exact test. Cox proportional hazards regression was used to obtain hazard ratios (HRs) of IBD and mortality among the matched cohorts. Kaplan-Meier curves were used to estimate the probability of autoimmune diseases onset or mortality, and the log-rank test was employed to analyze the differences between groups. All statistical analyses were performed using SAS 9.3 software (SAS Institute, Cary, NC). Statistical significance was set at $P < 0.05$.

## Results

### Higher incidence rates of autoimmune diseases and IBD in the PUD+HPRx in DM group as compared with the PUD-HPRx in general population and PUD-HPRx in DM groups

The incidence rates of autoimmune diseases and IBD in the three groups were compared. The baseline characteristics of the three groups (PUD-HPRx in general population, PUD-HPRx in DM, and PUD+HPRx in DM) are shown in Table 1. The average age and percentage of female subjects were 58.3 ± 14.50 years and 54.61% in the PUD-HPRx in general population group, 58.11 ± 14.12 years and 55.11% in the PUD+HPRx in DM group, and 58.23 ± 14.24 years and 55.10% in the PUD-HPRx in DM group.

Following multiple regression to adjust for age, sex, comorbidities, Charlson's comorbidities index score (CCIS) and medications, the results showed that the new incidence rates of autoimmune diseases and IBD were significantly higher in the PUD+HPRx in DM group than in the PUD-HPRx in general population and PUD-HPRx in DM groups (autoimmune diseases = 5.14% vs 3.47% and 3.65%; IBD = 5.60% vs 3.17% and 3.25%, respectively; $P < 0.0001$). The incidence rate of IBD with asacol and/or azathioprine therapy was also significantly higher in the PUD+HPRx in DM group than in the PUD-HPRx in general population and

**Table 1. Demographic data of PUD+HPRx and PUD-HPRx in DM and general population post-propensity score-matching.**

|  | PUD-HPRx in GP | PUD+HPRx in DM | PUD-HPRx in DM |  |
|---|---|---|---|---|
| N | 48,541 | 48,737 | 48,737 | p-value |
| Age in years (Mean±SD)* | 58.30 (±14.50) | 58.11 (±14.12) | 58.23 (±14.24) | 0.129 |
| Age categories (N, %) |  |  |  |  |
| <35 | 2,462 (5.07%) | 2,511 (5.15%) | 2,510 (5.15%) | 0.0613 |
| 35–44 | 5,682 (11.71%) | 5,817 (11.94%) | 5,815 (11.93%) |  |
| 45–54 | 11,599 (23.90%) | 11,800 (24.21%) | 11,802 (24.22%) |  |
| 55–64 | 11,652 (24.00%) | 11,898 (24.41%) | 11,898 (24.41%) |  |
| 65+ | 17,146 (35.32%) | 16,711 (34.29%) | 16,712 (34.29%) |  |
| Gender (N, %)* |  |  |  |  |
| Female | 26,509 (54.61%) | 26,857 (55.11%) | 26,854 (55.10%) | 0.2059 |
| Male | 22,032 (45.39%) | 21,880 (44.89%) | 21,883 (44.90%) |  |
| Income status (N, %)* |  |  |  |  |
| Dependent | 12,278 (25.29%) | 12,201 (25.03%) | 12,199 (25.03%) | 0.2106 |
| <20,000 | 12,799 (26.37%) | 12,645 (25.95%) | 12,647 (25.95%) |  |
| 20,000–40,000 | 16,920 (34.86%) | 17,099 (35.08%) | 17,100 (35.09%) |  |
| ≥ 40,000 | 6,544 (13.48%) | 6,792 (13.94%) | 6,791 (13.93%) |  |
| Comorbidities (N, %) |  |  |  |  |
| Hypertension * | 21,991 (45.30%) | 22,059 (45.26%) | 22,053 (45.25%) | 0.9838 |
| Hyperlipidemia* | 18,981 (39.10%) | 19,228 (39.45%) | 19,228 (39.45%) | 0.4362 |
| Myocardial infraction | 926 (1.91%) | 815 (1.67%) | 815 (1.67%) | 0.0054 |
| Congestive heart failure | 2,226 (4.59%) | 2,282 (4.68%) | 2,258 (4.63%) | 0.7741 |
| Peripheral vascular disease | 867 (1.79%) | 782 (1.60%) | 778 (1.60%) | 0.0326 |
| Cerebral vascular disease | 5,692 (11.73%) | 4,641 (9.52%) | 5,300 (10.87%) | < .0001 |
| Dementia | 992 (2.04%) | 806 (1.65%) | 919 (1.89%) | < .0001 |
| Chronic kideney disease | 3,020 (6.22%) | 2,548 (5.23%) | 2,633 (5.40%) | < .0001 |
| Cancer* | 3,228 (6.65%) | 3,248 (6.66%) | 3,238 (6.64%) | 0.9914 |
| Charlson's Index Score (Mean±SD) | 1.99 (±1.75) | 1.99 (±1.71) | 2.01 (±1.75) | 0.4738 |
| Charlson's Index Categories (N, %)* |  |  |  |  |
| 0 | 6,868 (14.15%) | 6,854 (14.06%) | 6,856 (14.07%) | 0.8841 |
| 1–2 | 16,036 (33.04%) | 15,973 (32.77%) | 15,974 (32.78%) |  |
| 3 | 12,116 (24.96%) | 12,157 (24.94%) | 12,155 (24.94%) |  |
| > = 4 | 13,521 (27.85%) | 13,753 (28.22%) | 13,752 (28.22%) |  |
| Medication (N, %) |  |  |  |  |
| Metformin | 18,461 (38.03%) | 43,147 (88.53%) | 41,014 (84.15%) | < .0001 |
| Sulfonylurea | 19,101 (39.35%) | 43,940 (90.16%) | 41,846 (85.86%) | < .0001 |
| DPP4 inhibitor | 1,850 (3.81%) | 5,704 (11.70%) | 4,913 (10.08%) | < .0001 |
| Insulin | 9,500 (19.57%) | 20,856 (42.79%) | 19,617 (40.25%) | < .0001 |
| NSAIDs* | 41,493 (85.48%) | 41,749 (85.66%) | 41,751 (85.67%) | 0.642 |
| Antiplatelet agent | 6,394 (13.17%) | 7,207 (14.79%) | 6,277 (12.88%) | < .0001 |
| Warfarin | 488 (1.01%) | 439 (0.90%) | 467 (0.96%) | 0.2438 |
| Protom pump inhibitor | 22,189 (45.71%) | 40,232 (82.55%) | 26,823 (55.04%) | < .0001 |
| H2-receptor antagonist | 38,746 (79.82%) | 46,259 (94.92%) | 40,514 (83.13%) | < .0001 |

Note:

*: variables used in the propensity score matching model.

Values are presented as n (%) or mean ± SD.

GP, general population; NSAID, nonsteroidal anti-inflammatory drug; PUD, peptic ulcer disease; PUD+HPRx, peptic ulcer disease with *Helicobacter pylori* treatment; PUD-HPRx, peptic ulcer disease without *Helicobacter pylori* treatment; DPP4 inhibitor, Dipeptidyl peptidase 4 inhibitor; H2-receptor antagonist, Histamine-2-receptor antagonist.

PUD-HPRx in DM groups (0.14% vs 0.05% and 0.06%) (Table 2 and Fig 2). The data of IBD was divided into Crohn's disease and chronic ulcerative colitis. Both the incidence of Crohn's disease and chronic ulcerative colitis were significantly increased in PUD+HPRx in DM group as that of IBD (Table 2).

## Lower all-cause mortality rate in the PUD+HPRx in DM group as compared with the PUD-HPRx in DM group

Multiple regression was performed to adjust for age, sex, comorbidities, CCIS, and medications in order to analyze the risk of death. The all-cause mortality was 17.20% in the PUD-HPRx in general population group, 19.98% in the PUD+HPRx in DM group, and 20.43% in the PUD-HPRx in DM group. Predictably, the PUD-HPRx in general population group had the lowest all-cause mortality, as compared with the PUD+HPRx in DM group (HR: 1.086; 95% confidence interval (CI), 1.054–1.118, P < 0.001) and the PUD-HPRx in DM group (HR: 1.159; 95%CI, 1.125–1.193, P < 0.001). Furthermore, a lower all-cause mortality

**Table 2. Autoimmune disease and inflammatory bowel disease outcomes in the matched cohorts.**

|  | General Population | DM Population | | |
|---|---|---|---|---|
|  | PUD-HPRx | PUD+HPRx | PUD-HPRx | P-value |
| N | 48,541 | 48,737 | 48,737 | |
| Two-year wash out period | | | | |
| Total follow-up person years (in years) | 324,794.24 | 344,539.06 | 334,239.47 | |
| Health outcomes (N,%) | | | | |
| autoimmune diseases | 1,511 (3.11%) | 2,253 (4.62%) | 1,560 (3.20%) | < .0001 |
| IBD | 1,302 (2.68%) | 2,349 (4.82%) | 1,338 (2.75%) | < .0001 |
| Crohn's disease | 1,094 (2.25%) | 2,481 (5.09%) | 1,174 (2.41%) | < .0001 |
| Chronic ulcerative colitis | 107 (0.22%) | 261 (0.54%) | 120 (0.25%) | < .0001 |
| Incidence rate ratio (95%CI) | | | | |
| IBD (ref. = PUD-HPRx in GP) | ref. | 1.701 (1.589,1.821)*** | 0.999 (0.925,1.079) | |
| IBD (ref. = PUD-HPRx in DM) | 1.001 (0.927,1.082) | 1.703 (1.592,1.823)*** | ref. | |
| Crohn's disease (ref. = PUD-HPRx in GP) | ref. | 2.138 (2.081,2.195)* | 1.043 (1.013,1.073)* | |
| Crohn's disease (ref. = PUD-HPRx in DM) | 0.959 (0.902,1.016) | 2.050 (2.020,2.080)* | ref. | |
| Chronic ulcerative colitis(ref. = PUD-HPRx in GP) | ref. | 2.299 (2.288,2.311)* | 1.090 (1.081,1.099)* | |
| Chronic ulcerative colitis(ref. = PUD-HPRx in DM) | 0.918 (0.906,0.929)* | 2.110 (2.101,2.119)* | ref. | |
| All-cause mortality | | | | |
| Total follow-up person years | 329,944.27 | 354,406.36 | 339,675.73 | |
| All-cause mortality | | | | |
| No | 40,193 (82.80%) | 39,000 (80.02%) | 38,779 (79.57%) | < .0001 |
| Yes | 8,348 (17.20%) | 9,737 (19.98%) | 9,958 (20.43%) | |
| Mortality rate ratio (95%CI) | | | | |
| Mortality (ref. = PUD-HPRx in GP) | ref. | 1.086 (1.054,1.118)*** | 1.159 (1.125,1.193)*** | |
| Mortality (ref. = PUD-HPRx in DM) | 0.863 (0.838,0.889)*** | 0.937 (0.911,0.964)*** | ref. | |

Note:

*: p<0.05

**: p<0.01

*** p<0.001.

Adjusted for age, sex, insurance range, comorbidities, Charlson comorbidity index score, and medications.

GP, general population; IBD, inflammatory bowel disease; CI, confidence interval.

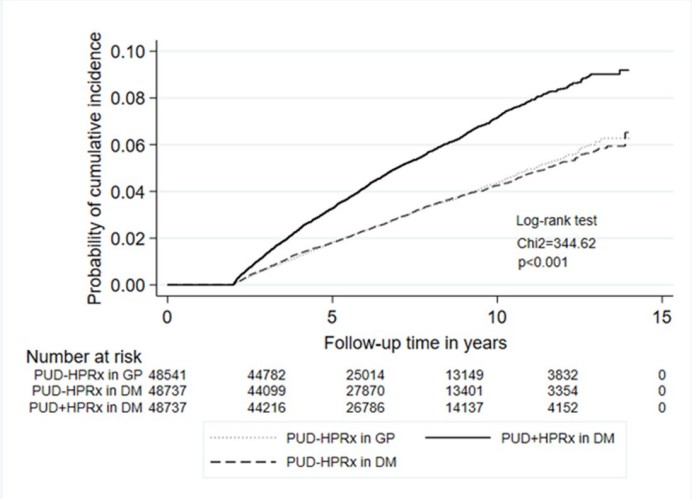

**Fig 2. Cumulative incidence of IBD with a two-year washout period.** Higher cumulative incidence of IBD in DM patients with peptic ulcer disease with *Helicobacter pylori* treatment (PUD+HPRx in DM) as compared with the general population with peptic ulcer disease without *Helicobacter pylori* treatment (PUD-HPRx in general population) and with DM patients with peptic ulcer disease without *Helicobacter pylori* treatment (PUD-HPRx in DM).

was noted in the PUD+HPRx in DM group (HR: 0.937; 95%CI, 0.911–0.964, P < 0.001) as compared with the PUD-HPRx in DM group (Table 2 and Fig 3).

## Higher incidence rate of IBD in younger patients in the PUD+HPRx in DM group as compared with the PUD-HPRx in DM group

Cox proportional hazards regression was performed on the outcomes of IBD and mortality among the matched cohorts. A trend of a higher incidence of IBD in the younger patients in

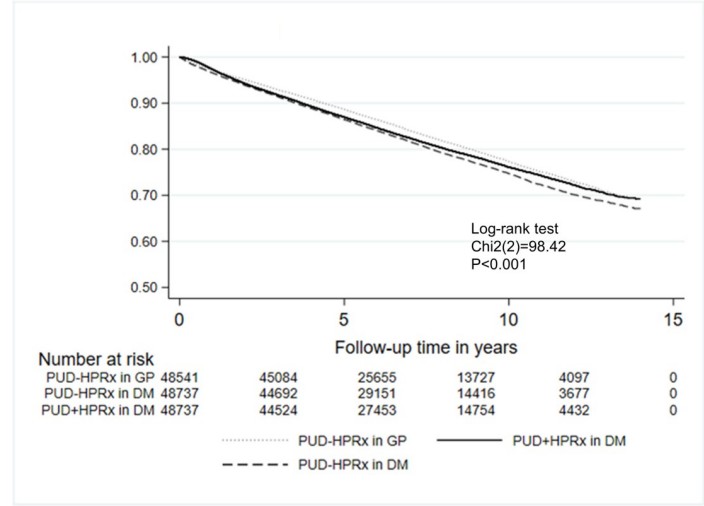

**Fig 3. Cumulative survival rate.** Lower cumulative survival rate in DM patients with peptic ulcer disease without *Helicobacter pylori* treatment (PUD-HPRx in DM) as compared with DM patients with peptic ulcer disease with *Helicobacter pylori* treatment (PUD+HPRx in DM) and with the general population with peptic ulcer disease without *Helicobacter pylori* treatment (PUD-HPRx in general population). Kaplan-Meier curves estimated the probability of survival rate.

the PUD+HPRx in DM group as compared with the PUD-HPRx in DM group was observed
(< 35 years: HR: 1.805; 35–44 years: HR: 1.726; 45–54 years: HR: 1.603; 55–64 years: HR:
1.606; >65 years: HR: 1.56) (Table 3 and Fig 4).

## Younger patients had a lower mortality rate in the PUD+HPRx in DM group as compared with the PUD-HPRx in DM group

A trend of a lower mortality rate in the younger patients in the PUD+HPRx in DM group as
compared with the PUD-HPRx in DM group was observed (<35 years: HR: 0.829; 35–44
years: HR: 0.877; 45–54 years: HR: 0.937; 55–64 years: HR: 0.991; > 65 years: HR: 0.938)
(Table 3 and Fig 4).

## Discussion

Our results revealed that the incidences of autoimmune diseases and IBD in the PUD+HPRx
in DM group were higher than those in the PUD-HPRx in general population and PUD-HPRx
in DM groups. This finding suggested that treatment for *H. pylori* infection can increase the
risk of autoimmune diseases and IBD in patients with DM, and not only in the general
population.

   In a study by Lin et al., the authors demonstrated that treatment for *H. pylori* infection in
the general population is associated with significant increases in the risks of autoimmune dis-
eases and IBD as compared with people who had never received *H. pylori* therapy. According
to the ACG Clinical Guidelines: Treatment of *H. pylori* Infection (2017) [18], currently, the
indications to test for *H. pylori* infection are to examine patients with PUD or a history of
PUD, low-grade gastric mucosa-associated lymphoid tissue (MALT) lymphoma, or a history
of endoscopic resection of early gastric cancer. If the result is positive, then treatment should
be performed. However, a recent meta-analysis of 24 studies (mostly in Asia) showed that *H.
pylori* treatment in asymptomatic infected adults had reduced the incidence of gastric cancer.
The greatest benefit was found in the area with the highest incidence of gastric cancer [19].
Widely eradication of *H. pylori* may be a trend in the future.

   According to the literature, *H. pylori* infection may increase several risk factors in diabetic
patients. A meta-analysis in 2013 identified relationships between *H. pylori* infection and
increased risks of nephropathy and neuropathy [20]. It has been reported that *H. pylori* is asso-
ciated with poor control of blood sugar in type 1 DM patients [21]. In addition, *H. pylori* infec-
tion may increase the risk of cardiovascular disease in patients with coronary heart disease and
type 2 DM [22].

   Novel information regarding gut microbiota may explain the effects associated with *H.
pylori* infection. We have long been aware that imbalance of intestinal microbiota plays a role
in the development of several diseases, including DM and IBD [23, 24]. A recent study revealed
that modification of microbiota can repair the structure and function of the intestinal barrier
in diabetic mice [25]. Clinical application of microbial therapy is under investigation. On the
other hand, *H. pylori* infection was found to cause peptic ulcers and gastric cancer, but acts
against several autoimmune diseases. Therefore, it has been suggested that *H. pylori* has both
protective and pathogenic effects, and thus the interaction between microbiota and host may
influence autoimmunity and the inflammatory response [26].

   The results of our study also revealed that the earlier an *H. pylori* infection is treated, the
higher the risk of developing IBD (Fig 4); the exact mechanism remains unclear. Study showed
that the composition of intestinal microbiota changes with age and disease [27]. Our hypothe-
sis was that the earlier the change in microbiota, the greater the influence on the host response.
Nevertheless, a lower all-cause mortality rate was found in the PUD+HPRx in DM group as

**Table 3. Cox proportional hazards model assessment of outcomes of inflammatory bowel disease and mortality in the matched cohorts.**

| | IBD incidence with two year washout period | Mortality |
|---|---|---|
| **Variables** | **HR (95%CI)** | **HR (95%CI)** |
| Cohorts (ref. group: PUD-HPRx in DM) | | |
| PUD-HPRx in GP | 0.999 (0.925,1.078) | 0.794 (0.771,0.817)*** |
| PUD+HPRx in DM | 1.679 (1.570,1.795)*** | 0.930 (0.904,0.956)*** |
| Age categories (ref. group: <35)# | | |
| 35–44 | 0.850 (0.749,0.965)* | 1.489 (1.324,1.676)*** |
| 45–54 | 0.825 (0.733,0.928)** | 1.730 (1.549,1.932)*** |
| 55–64 | 0.805 (0.713,0.909)** | 2.362 (2.118,2.633)*** |
| 65+ | 0.811 (0.717,0.917)** | 6.068 (5.455,6.750)*** |
| Gender (ref. female) | | |
| Male | 0.827 (0.780,0.878)*** | 1.623 (1.583,1.663)*** |
| Income status (ref.: dependent) | | |
| <20,000 | 1.039 (0.958,1.126) | 1.254 (1.218,1.292)*** |
| 20,000–40,000 | 0.995 (0.924,1.072) | 0.825 (0.800,0.851)*** |
| 40,000 | 0.938 (0.849,1.037) | 0.372 (0.347,0.399)*** |
| Comorbidities (N, %) | | |
| Hypertension (ref.: No) | 0.964 (0.905,1.027) | 0.966 (0.941,0.991)*** |
| Hyperlipidemia (ref.:No) | 0.941 (0.883,1.002) | 1.319 (1.286,1.352)*** |
| Myocardial infraction (ref.: No) | 0.946 (0.733,1.221) | 1.107 (1.041,1.178)** |
| Congestive heart failure (ref.: No) | 1.164 (1.002,1.352)* | 1.603 (1.543,1.665)*** |
| Peripheral vascular disease (ref.:No) | 1.160 (0.932,1.443) | 1.100 (1.023,1.183)* |
| Cerebral vascular disease (ref.:No) | 0.914 (0.816,1.023) | 1.302 (1.261,1.344)*** |
| Dementia (ref.:No) | 0.909 (0.672,1.229) | 1.939 (1.842,2.040)*** |
| Chronic kidney disease (ref.:No) | 0.938 (0.809,1.088) | 1.487 (1.431,1.546)*** |
| Cancer (ref.:No) | 0.973 (0.843,1.123) | 1.934 (1.865,2.005)*** |
| Charlson's Index Categories (ref.: CCI = 0) | | |
| 1–2 | 1.015 (0.931,1.106) | 1.103 (1.045,1.164)** |
| 3 | 1.061 (0.967,1.165) | 1.324 (1.255,1.398)*** |
| > = 4 | 1.084 (0.972,1.209) | 1.799 (1.702,1.902)*** |
| Medication (N, %) | | |
| Metformin (ref.: No) | 1.159 (1.099,1.222)*** | 1.144 (1.116,1.173)*** |
| Sulfonylurea (ref.: No) | 1.194 (1.131,1.260)*** | 1.343 (1.308,1.378)*** |
| Dipeptidyl peptidase 4 inhibitor (ref.: No) | 1.081 (0.993,1.176) | 0.653 (0.622,0.687)*** |
| Insulin (ref.: No) | 1.093 (1.038,1.151)*** | 3.714 (3.628,3.801)*** |
| NSAIDs (ref.: No) | 1.426 (1.298,1.568)*** | 0.816 (0.790,0.843)*** |
| Antiplatelet agent (ref.: No) | 1.141 (1.046,1.245)** | 1.014 (0.984,1.044) |
| Warfarin (ref.: No) | 0.917 (0.642,1.309) | 1.242 (1.144,1.349)*** |

Note:

*: p<0.05

**: p<0.01

*** p<0.001.

#Likelihood-ratio test for trend for the age categories were significant for model 1 (LR chisq = 12.73, p = 0.0053), model 2 (LR chisq = 11.42, p = 0.097) and model 3 (LR chisq = 1454.76, p<0.001), indicating a linear trend across age groups for the dependent variables.

Adjusted for age, sex, insurance range, comorbidities, Charlson comorbidity index score, and medications.

HR, hazard ratio; CI, confidence interval; GP, general population; IBD, inflammatory bowel disease; ref.: No, reference: the cohort without the comorbidity or medication.

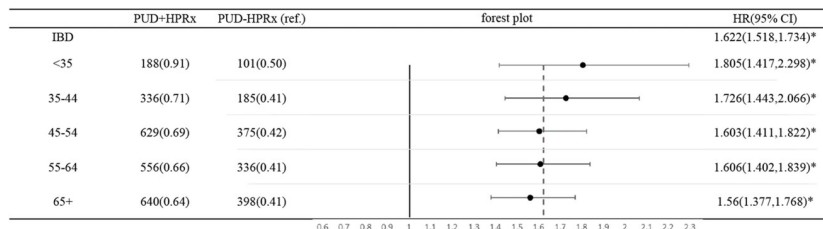

| | PUD+HPRx | PUD-HPRx (ref.) | forest plot | HR(95% CI) |
|---|---|---|---|---|
| IBD | | | | 1.622(1.518,1.734)* |
| <35 | 188(0.91) | 101(0.50) | | 1.805(1.417,2.298)* |
| 35-44 | 336(0.71) | 185(0.41) | | 1.726(1.443,2.066)* |
| 45-54 | 629(0.69) | 375(0.42) | | 1.603(1.411,1.822)* |
| 55-64 | 556(0.66) | 336(0.41) | | 1.606(1.402,1.839)* |
| 65+ | 640(0.64) | 398(0.41) | | 1.56(1.377,1.768)* |

| | PUD+HPRx# | PUD-HPRx (ref.)# | Forest plot (Xaxis:HR) | HR(95% CI) |
|---|---|---|---|---|
| Death | | | | 0.937(0.911,0.963)* |
| <35 | 134(0.62) | 154(0.75) | | 0.829(0.658,1.045) |
| 35-44 | 452(0.92) | 486(1.05) | | 0.877(0.772,0.997)* |
| 45-54 | 1022(1.08) | 1042(1.16) | | 0.937(0.859,1.021) |
| 55-64 | 1571(1.81) | 1539(1.83) | | 0.991(0.924,1.063) |
| 65+ | 6558(6.38) | 6753(6.81) | | 0.938(0.907,0.971)* |

# no. of events(rate/100person-year)
* P-value<0.05

**Fig 4. Forest plots of the PUD+HPRx and PUD-HPRx in DM groups.** A higher hazard ratio of the incidence rate of IBD was noted in the younger patients in the PUD+HPRx in DM group, while a lower hazard ratio of the mortality rate was noted in the younger patients in the PUD+HPRx in DM group.

compared with the PUD-HPRx in DM group, and younger patients had a lower mortality in the PUD+HPRx in DM group as compared with the PUD-HPRx in DM group. The exact causes of death in these patients were unclear owing to limitations of the database. However, we speculated that treatment of *H. pylori* decreases the risks of gastric cancer and cardiovascular disease, which are increased in DM patients.

Congestive heart failure and use of anti-platelet agent are also risk factors for IBD incidence with two-year washout period. We speculate that using anti-platelet agent increase the risk of peptic ulcer, and more patients with peptic ulcer received EGD and H. pylori biopsy. Patients with congestive heart failure frequently use anti-platelet agent may lead to the same result.

On the other hand, Table 3 showed using metformin, sulfonylurea, and insulin increased the incidence of IBD, but DPP4 inhibitor did not. We performed the subgroup analysis to compare the incidence of IBD in patients with and without the specific DM medication in PUD+HPRx in DM and PUD-HPRx in DM group (see S3 Table). The result showed DM medications didn't increase the incidence of IBD.

There were some limitations in this observational study. First, the report of H. pylori biopsy do not present in the database. In Taiwan, H. pylori biopsy will be performed during EGD. If H. pylori infection is confirmed, H. pyloritherapy will be given. This limitation makes us unable to exclude the rare patients with H. pylori infection but refused or unable to receive H. pylori treatment. But it doesn't influence our result. Because the patients with H. pylori infected almost always receive treatment in Taiwan, the PUD patients not received H. pylori treatment may represent negative results of H. pylori infection. Secondary, some patients received H. pylori therapy before 2000 might be grouped to PUD-HPRx because our patients were from 2000 to 2010. This doesn't influence the result because these patients actually with higher risk of IBD were grouped to PUD-HPRx, which has lower risk of IBD. Third, the sample of PUD patients who were treated for H. pylori infection in general population (PUD +HPRx in general population) was too small to match to other three groups. However, our (S1 and S2 Tables) including PUD+HPRx in general population group revealed similar result. Fourth, the result of H. pylori treatment can't be presented in the database. The success rate of H. pylori treatment in Taiwan was 82–94% under 7 days triple therapy [28], and 84% under

third-line therapy [29]. Last, the severity of IBD was defined by the clinical signs such as frequency of diarrhea, abdominal pain, fever, anemia. . .etc, which were not included in our database.

In conclusion, our results revealed that treatment of *H. pylori* infection increases the incidence rates of autoimmune diseases and IBD, and decreases the all-cause mortality rate in the DM group with PUD; in addition, the effect was more significant in younger patients. This finding may increase the understanding of clinicians of the possible risks and benefits of *H. pylori* therapy in patients with DM. Besides, widely treatment of *H. pylori* may cause unexpected effects such as autoimmune diseases and IBD. Further associated studies should be performed.

## Supporting information

**S1 Table. Demographic prevalence of PUD+HPRx and PUD-HPRx in a DM population and a general population post-propensity score-matching.**
(DOC)

**S2 Table. Autoimmune disease and inflammatory bowel disease outcomes in the matched cohorts.**
(DOC)

**S3 Table. Cox proportion model results for outcomes of IBD and mortality among matched cohorts.**
(DOC)

## Acknowledgments

The authors thank the help from the Division of Medical Statistics and Bioinformatics, Department of Medical Research, Kaohsiung Medical University Hospital, and Center for Big Data Research (KMUHD-108023), Kaohsiung Medical University for providing administrative support, including the Kaohsiung Medical University Hospital Research Database (KMUHRD) and Taiwan Liver Research Foundation for their assistance.

## Author Contributions

**Conceptualization:** Deng-Chyang Wu, John Y. Kao, Kun-Der Lin.

**Data curation:** Nai-Wei Sheu, Shu-Heng Huang.

**Writing – original draft:** Nai-Wei Sheu.

**Writing – review & editing:** John Y. Kao, Kun-Der Lin.

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
