## [Decision Letter · Decision Letter 0]

10 Aug 2021

PONE-D-21-21394

Effects of Helicobacter pylori eradication on the incidences of autoimmune diseases and inflammatory bowel disease in patients with diabetes mellitus

PLOS ONE

Dear Dr. Kun-Der Lin,

Thank you for submitting your manuscript to PLOS ONE. After careful consideration, we feel that it has merit but does not fully meet PLOS ONE’s publication criteria as it currently stands. Therefore, we invite you to submit a revised version of the manuscript that addresses the points raised during the review process.

This is important study for investigate the relationship between AD and DM etc., and have several new findings. However as reviewers also pointed out, there are several concerns in this study (please especially check the comments by Reviewer 1).  To accept this paper, you should carefully respond each comment by the reviewers. Discussion should have more information about the study. 

We look forward to receiving your revised manuscript.

Kind regards,

Yoshio Yamaoka

Academic Editor

PLOS ONE

Journal Requirements:

2. This is a retrospective study with no control group. As such, we do not feel that any conclusions on the intervention effects can be supported; as such, we ask that you revise the text (especially, but no limited to, the aims and Conclusions) to avoid unsupported statements.

“This study was funded by a grant from the Ministry of Health and Welfare (MOHW108-TDU-B-212-133006), Kaohsiung Medical University (109CM-KMU- 012), Kaohsiung Medical University Hospital (SA10905), and Kaohsiung Municipal Ta-Tung Hospital (KMTTH-107-014), which had no role in the study design, data analysis, data interpretation, or writing of the manuscript.”

Reviewers' comments:

Reviewer's Responses to Questions

**Comments to the Author**

1. Is the manuscript technically sound, and do the data support the conclusions?

Reviewer #1: No

Reviewer #2: Partly

2. Has the statistical analysis been performed appropriately and rigorously? 

Reviewer #1: I Don't Know

Reviewer #2: Yes

3. Have the authors made all data underlying the findings in their manuscript fully available?

Reviewer #1: Yes

Reviewer #2: Yes

4. Is the manuscript presented in an intelligible fashion and written in standard English?

Reviewer #1: No

Reviewer #2: Yes

5. Review Comments to the Author

Reviewer #1: In this paper, Dr. Sheu et al. performed an observational study using National Health Insurance Research Database in Taiwan. The authors found that H. pylori treatment increased the incidences of autoimmune diseases (AD) and IBDs and decreased the all-cause mortality in the diabetes patients with peptic ulcer disease (PUD). Although the results are interesting, several questions should be addressed to accept the association is a true association.

Major comments

1. Due to the retrospective observation study design, the findings are vulnerable to many biases. Why DM patients were selected as the main study group? Because one of control group was PUD patients not treated for H. pylori in general population (PUD-HpRx in GP), PUD patients who were treated for Hp infection in general population might be better to assess the Hp treatment and AD or IBD association.

2. PUD is a well-known indication for H. pylori treatment since 1990s. Why so many PUD patients in this database had not treated for H. pylori? Especially in DM group, the number who did not receive H. pylori treatment seems more than 10 times in number. This might raise a question about the completeness or correctness of the database.

3. The persons in the control groups who were not treated for H. pylori in general population or in DM patients should be very heterogenous. The persons are one of the cases; 1) H. pylori uninfected persons, 2) H. pylori infected but not treated.

4. The persons in the case group who were treated for H. pylori also seems very heterogenous. Those are one of the cases; 1) H. pylori treatment medication was prescribed but not took at all (poor compliance), 2) H. pylori treatment medication was took, but H. pylori eradication was either successful or failed. Since the eradication success data are not available from the database, the title and description needs be modified to more modest term such as H. pylori treatment not H. pylori eradication. Actually it’s very hard to tell whether exposure to H. pylori treatment as whole, H. pylori eradication itself, or just an exposure of certain antibiotic included in the H. pylori treatment is the main culprit for the authors finding.

5. One of main finding was that H. pylori treatment decreased the all-cause mortality in the DM group with PUD. This finding might also be misleading because there is a chance that the patients having more severe DM were not treated for H. pylori infection. If mild DM patients were more selectively treated for H. pylori, the group should have better survival rates.

6. Please clarify the study entry date and end date. In line 140 on page 6, the persons who had received treatment for H. pylori infection between 1997 and 2013 were enrolled. In line 148 on page 6, the sentence was described as “The person-years of follow-up were estimated from the index date plus a two-year washout duration to the date of diagnosis of AD or IBD, or an end date of December 31, 2010.” The sentence might be interpreted as the end date of follow up of this study was on Dec 31, 2010, which is much earlier than 2013.

7. Discussion seems too short and more plausible link between IBD, AD and mortality needs be provided. Also, the limitations of this study should be described in detail.

Reviewer #2: General:

In this study, the authors investigated effects of H. pylori eradication therapy in patients with diabetes mellitus (DM). Authors showed that higher incidences of autoimmune diseases and inflammatory bowel disease (IBD) were observed in the DM patients with peptic ulcer and received eradication therapy than in the general population with peptic ulcer and received eradication therapy and DM patients with peptic ulcer disease and non-eradication therapy. Authors concluded that H. pylori eradication therapy increased the incidences of autoimmune diseases and IBD and decreased the all-cause mortality in the DM group with peptic ulcer.

This study was well written.

Major comments:

1. Authors selected three groups. However, Authors should add one group: general population with peptic ulcer who received H. pylori treatment.

2. In this study, authors included patients with peptic ulcer. If authors enrolled patients with gastritis, not peptic ulcer, could the results be different?

3. Although authors enrolled many patients, direct association with autoimmune diseases, inflammatory bowel disease and diabetes mellitus is unclear.

4. Why did authors select patients with diabetes mellitus?

5. H. pylori treatment decreased the all-cause mortality in the DM group with peptic ulcer. Why?

6. What is the relationship between the time received H. pylori eradication therapy and the time of autoimmune diseases and IBD onset?

7. Congestive heart failure and use of anti-platelet agent are risk factor for IBD incidence with two-year washout period. Why?

8. Authors should add data of intake, kinds and dose of medication for DM.

9. Please add p values in Table 1.

10. What is a p value in Table 2? PUD+HPRx vs PUD-HPRx?

11. Several H. pylori bacterial virulence factors that dampen the immune response of the host were identified, including VacA, CagA, and H. pylori lipopolysaccharide. However, Western population infected with cagA-negative and vacA s2m2 type H. pylori. Therefore, influence of eradication therapy for IBD may differ between Asian population and Western population.

12. Authors used data received treatment for H. pylori infection between 1997 and 2013. How is the judgment for eradication therapy?

13. How about H. pylori infection rate in a group with peptic ulcer without H. pylori treatment?

6. PLOS authors have the option to publish the peer review history of their article (what does this mean?). If published, this will include your full peer review and any attached files.

Reviewer #1: No

Reviewer #2: No

---

## [Author Response · Author response to Decision Letter 0]

23 Sep 2021

Editor’s comment: 

Comment 1:

Answer: 

We had modified our manuscript to meet the PLOS ONE's style requirements, and renamed our figures according to the cited order in manuscript. Thanks for your kind reminder. 

Comment 2:

This is a retrospective study with no control group. As such, we do not feel that any conclusions on the intervention effects can be supported; as such, we ask that you revise the text (especially, but no limited to, the aims and Conclusions) to avoid unsupported statements. 

Answer: 

Our control groups were PUD-HPRx in DM and PUD-HPRx in GP group. We compared the incidence of IBD and mortality of PUD+HPRx in DM group to other two groups. Besides, we presented the result included PUD+HPRx in GP group in the supporting tables.

Comment 3:

Thank you for stating the following in the Acknowledgments Section of your manuscript:“This study was funded by a grant from the Ministry of Health and Welfare (MOHW108-TDU-B-212-133006), Kaohsiung Medical University (109CM-KMU- 012), Kaohsiung Medical University Hospital (SA10905), and Kaohsiung Municipal Ta-Tung Hospital (KMTTH-107-014), which had no role in the study design, data analysis, data interpretation, or writing of the manuscript.”

Please remove any funding-related text from the manuscript and let us know how you would like to update your Funding Statement. Currently, your Funding Statement reads as follows:“The author(s) received no specific funding for this work.”

Answer: 

Thank you for your suggestion. We would remove the funding-related text from the manuscript. Please change our Funding Statement to 

“This study was funded by a grant from the Ministry of Health and Welfare (MOHW108-TDU-B-212-133006), Kaohsiung Medical University (109CM-KMU- 012), Kaohsiung Medical University Hospital (SA10905), and Kaohsiung Municipal Ta-Tung Hospital (KMTTH-107-014), which had no role in the study design, data analysis, data interpretation, or writing of the manuscript. ”

Reviewer 1:

Comment 1:

Due to the retrospective observation study design, the findings are vulnerable to many biases. Why DM patients were selected as the main study group? Because one of control group was PUD patients not treated for H. pylori in general population (PUD-HpRx in GP), PUD patients who were treated for Hp infection in general population might be better to assess the Hp treatment and AD or IBD association.

Answer: 

We selected DM patients as the main study group because DM and PUD are common disease in Taiwan. The data of DM patients were more intact in our database. The supporting table include PUD patients who were treated for Hp infection in general population (PUD+HpRx in GP). The result was similar. The sample of PUD+HpRx in GP group was too small to match to other groups.

Comment 2:

PUD is a well-known indication for H. pylori treatment since 1990s. Why so many PUD patients in this database had not treated for H. pylori? Especially in DM group, the number who did not receive H. pylori treatment seems more than 10 times in number. This might raise a question about the completeness or correctness of the database.

Answer: 

In Taiwan, H. pylori biopsy will be performed during Esophagogastroduodenoscopy (EGD). If H. pylori infection is confirmed, H. pylori eradication will be given. But the report of H. pylori biopsy do not present in the database. This limitation makes us unable to exclude the rare patients with H. pylori infection but refused or unable to receive H. pylori treatment. But it doesn’t influence our result. Due to the patients with H. pylori infected almost always receive treatment in Taiwan, the PUD patients not received H. pylori treatment may represent negative results of H. pylori infection.

Comment 3:

The persons in the control groups who were not treated for H. pylori in general population or in DM patients should be very heterogenous. The persons are one of the cases; 1) H. pylori uninfected persons, 2) H. pylori infected but not treated.

Answer: 

Yes. The condition(H. pylori infected but not treated) is rare in Taiwan. Due to the patients with H. pylori infected almost always receive treatment in Taiwan, the PUD patients not received H. pylori treatment may represent negative results of H. pylori infection.

Comment 4:

The persons in the case group who were treated for H. pylori also seems very heterogenous. Those are one of the cases; 1) H. pylori treatment medication was prescribed but not took at all (poor compliance), 2) H. pylori treatment medication was took, but H. pylori eradication was either successful or failed. Since the eradication success data are not available from the database, the title and description needs be modified to more modest term such as H. pylori treatment not H. pylori eradication. Actually it’s very hard to tell whether exposure to H. pylori treatment as whole, H. pylori eradication itself, or just an exposure of certain antibiotic included in the H. pylori treatment is the main culprit for the authors finding.

Answer: 

Yes. The result of H. pylori treatment can’t be presented in the database. It’s one of the limitation in our study. The success rate of H. pylori treatment in Taiwan was 82-94% under 7 days triple therapy, and 84% under third-line therapy. We had modified at Discussion section, paragraph 7, line 12~13. We will modify the term H. pylori eradication to H. pylori treatment. Thanks for your suggestion.

We define the patients with H. pylori treatment by the set of triple or quadruple therapy for more than 7 days as our previous study by Lin et al. (Clin Gastroenterol Hepatol. 2019 Sep;17(10):1991-1999) The therapy include proton pump inhibitor twice daily use, which differ from antibiotic therapy for other infection.

Comment 5:

One of main finding was that H. pylori treatment decreased the all-cause mortality in the DM group with PUD. This finding might also be misleading because there is a chance that the patients having more severe DM were not treated for H. pylori infection. If mild DM patients were more selectively treated for H. pylori, the group should have better survival rates.

Answer: 

The condition(H. pylori infected but not treated) is rare in Taiwan. The rare condition may happen in Intensive Care Unit when the patient is critical. However, our database was from Outpatient department where patients were stable. And the result had been adjusted by Charlson Comorbidity Index score.

Comment 6:

Please clarify the study entry date and end date. In line 140 on page 6, the persons who had received treatment for H. pylori infection between 1997 and 2013 were enrolled. In line 148 on page 6, the sentence was described as “The person-years of follow-up were estimated from the index date plus a two-year washout duration to the date of diagnosis of AD or IBD, or an end date of December 31, 2010.” The sentence might be interpreted as the end date of follow up of this study was on Dec 31, 2010, which is much earlier than 2013.

Answer: 

The database is 1997-2013. We enrolled patients from 2000-2010 for more intact data. We had modified in line 136~137 on page 6.

Comment 7:

Discussion seems too short and more plausible link between IBD, AD and mortality needs be provided. Also, the limitations of this study should be described in detail.

Answer: 

Thanks for your suggestion. We will revise the discussion section and limitations more completely.

Reviewer 2:

Comment 1:

Authors selected three groups. However, Authors should add one group: general population with peptic ulcer who received H. pylori treatment.

Answer: 

The supporting table include PUD patients who were treated for Hp infection in general population (PUD+HpRx in GP). The result was similar. The sample of PUD+HpRx in GP group was too small to match to other groups.

Comment 2:

In this study, authors included patients with peptic ulcer. If authors enrolled patients with gastritis, not peptic ulcer, could the results be different?

Answer: 

The coding of gastritis is not accurate because many patients with gastritis usually take oral medicine without received Esophagogastroduodenoscopy (EGD). However, patients with peptic ulcer were almost received EGD. So the coding of peptic ulcer is more accurate than gastritis.

Comment 3:

Although authors enrolled many patients, direct association with autoimmune diseases, inflammatory bowel disease and diabetes mellitus is unclear.

Answer: 

Our results revealed that treatment of H. pylori infection was associated with the increased incidence rates of ADs and IBD in DM patients with PUD. The incidence rate of IBD in PUD-HPRx in DM group did not increase compared to PUD-HPRx in GP group. However, further studies should be performed.

Comment 4:

Why did authors select patients with diabetes mellitus?

Answer: 

We selected DM patients as the main study group because DM and PUD are common disease in Taiwan. And the data of DM patients were more intact in our database. The supporting table include PUD patients who were treated for Hp infection in general population (PUD+HpRx in GP). The result was similar. The sample of PUD+HpRx in GP group was too small to match to other groups.

Comment 5:

H. pylori treatment decreased the all-cause mortality in the DM group with peptic ulcer. Why?

Answer: 

H. pylori treatment also decreased the all-cause mortality in the GP group with peptic ulcer (see the supporting table). We speculated that treatment of H. pylori decreases the risks of gastric cancer and cardiovascular disease, which are increased in DM patients.

Comment 6:

What is the relationship between the time received H. pylori eradication therapy and the time of autoimmune diseases and IBD onset?

Answer: 

The person-years of follow-up were estimated from the index date plus a two-year washout duration to the date of diagnosis of AD or IBD, or an end date. The H. pylori therapy may change intestinal microbiota, which may induce the incidence of AD or IBD.

Comment 7:

Congestive heart failure and use of anti-platelet agent are risk factor for IBD incidence with two-year washout period. Why?

Answer: 

We speculate that using anti-platelet agent increase the risk of peptic ulcer, and more patients with peptic ulcer received EGD and H. pylori biopsy. Patients with congestive heart failure frequently use anti-platelet agent may lead to the same result. We added our speculation at Discussion section, paragraph 6. Thanks for your question.

Comment 8:

Authors should add data of intake, kinds and dose of medication for DM.

Answer: 

The data of intake was not included in our database. We added the data of medications for DM in the table 1 and 3. Table 3 showed using metformin, sulfonylurea, and insulin increased the incidence of IBD, but DPP4 inhibitor did not. We performed the subgroup analysis to compare the incidence of IBD in patients with and without the specific DM medication in PUD+HPRx in DM and PUD-HPRx in DM group(see supporting table 2). The result showed DM medications didn’t increase the incidence of IBD. We added this finding at Discussion section, paragraph 7.

Comment 9:

Please add p values in Table 1.

Answer: 

We will correct the table in the manuscript. Thanks for your kind reminder.

Comment 10:

What is a p value in Table 2? PUD+HPRx vs PUD-HPRx?

Answer: 

We will correct the table in the manuscript. Thanks for your kind reminder.

Comment 11:

Several H. pylori bacterial virulence factors that dampen the immune response of the host were identified, including VacA, CagA, and H. pylori lipopolysaccharide. However, Western population infected with cagA-negative and vacA s2m2 type H. pylori. Therefore, influence of eradication therapy for IBD may differ between Asian population and Western population.

Answer: 

Yes, the different types of H. pylori infection between Asian and Western population may influence the result of effect of H. pylori therapy for IBD. The types of H. pylori were not included in our database. It needs more study include Western population for provement.

Comment 12:

Authors used data received treatment for H. pylori infection between 1997 and 2013. How is the judgment for eradication therapy?

Answer: 

The database is 1997-2013. We enrolled patients from 2000-2010 for more intact data. We had modified it in line 139~140 on page 6. Patients received H. pylori therapy before 2000 were grouped to PUD-HPRx. This doesn’t influence the result because these patients actually with higher risk of IBD were grouped to PUD-HPRx, which has lower risk of IBD.

Comment 13:

How about H. pylori infection rate in a group with peptic ulcer without H. pylori treatment?

Answer: 

The condition (H. pylori infected but not treated) is rare in Taiwan. Due to the patients with H. pylori infection almost always receive treatment in Taiwan, the PUD patients not receive H. pylori treatment may represent negative results of H. pylori infection.

---

## [Decision Letter · Decision Letter 1]

14 Dec 2021

PONE-D-21-21394R1Effects of Helicobacter pylori treatment on the incidences of autoimmune diseases and inflammatory bowel disease in patients with diabetes mellitusPLOS ONE

Dear Dr. Kun-Der Lin,

Thank you for submitting your manuscript to PLOS ONE. After careful consideration, we feel that it has merit but does not fully meet PLOS ONE’s publication criteria as it currently stands. Therefore, we invite you to submit a revised version of the manuscript that addresses the points raised during the review process. The revised version is now improved; however there are still several concerns as pointed out by the reviewer. Please fix the manuscript according to the comments.

We look forward to receiving your revised manuscript.

Kind regards,

Yoshio Yamaoka

Academic Editor

PLOS ONE

Reviewers' comments:

Reviewer's Responses to Questions

**Comments to the Author**

1. If the authors have adequately addressed your comments raised in a previous round of review and you feel that this manuscript is now acceptable for publication, you may indicate that here to bypass the “Comments to the Author” section, enter your conflict of interest statement in the “Confidential to Editor” section, and submit your "Accept" recommendation.

Reviewer #2: All comments have been addressed

2. Is the manuscript technically sound, and do the data support the conclusions?

Reviewer #2: Yes

3. Has the statistical analysis been performed appropriately and rigorously? 

Reviewer #2: Yes

4. Have the authors made all data underlying the findings in their manuscript fully available?

Reviewer #2: Yes

5. Is the manuscript presented in an intelligible fashion and written in standard English?

Reviewer #2: Yes

6. Review Comments to the Author

Reviewer #2: Authors revised according to reviewer’s suggestion and recommendation.

However, additional revision will be required.

1. Many abbreviations were used. Popular abbreviations will be acceptable, such as DM, IBD and PUD. However, minor abbreviations should be avoided to use, such as AD and GP.

2. IBD included Crohn’s disease and chronic ulcerative colitis. Pathogenesis of Crohn’s disease and chronic ulcerative colitis differ. Please show divide data into Crohn’s disease and chronic ulcerative colitis.

3. How about severity of IBD?

4. Please add data of PPI/H2RA in Table.

7. PLOS authors have the option to publish the peer review history of their article (what does this mean?). If published, this will include your full peer review and any attached files.

Reviewer #2: No

---

## [Author Response · Author response to Decision Letter 1]

23 Jan 2022

Dear Professor Yoshio Yamaoka:

 I sincerely thank you for your processing our manuscript “PONE-D-21-21394

Effects of Helicobacter pylori eradication on the incidences of autoimmune diseases and inflammatory bowel disease in patients with diabetes mellitus” and for your decision. We also would like to express our cordial appreciation to reviewers for their excellent suggestions. As reviewer’s suggestion, we created supporting tables. Additionally, we have carefully revised our manuscript according to your comments. Our point-by-point responses are listed below.

Reviewer 2:

Comment 1:

Many abbreviations were used. Popular abbreviations will be acceptable, such as DM, IBD and PUD. However, minor abbreviations should be avoided to use, such as AD and GP.

Answer: 

We had modified our manuscript to avoid use minor abbreviations such as AD and GP. Thanks for your kind reminder. 

Comment 2:

IBD included Crohn’s disease and chronic ulcerative colitis. Pathogenesis of Crohn’s disease and chronic ulcerative colitis differ. Please show divide data into Crohn’s disease and chronic ulcerative colitis.

Answer: 

We had divided the data of IBD to Crohn’s disease and chronic ulcerative colitis (see Table 2). Both the incidence of Crohn’s disease and chronic ulcerative colitis were significantly increased in PUD+HPRx in DM group as that of IBD. We had modified at Results section, paragraph 1, line 12~15 on page 8. Cox proportional hazards regression was performed on the outcomes of Crohn’s disease and chronic ulcerative colitis among the matched cohorts (Table S4) and the results were similar. Because the sample of chronic ulcerative colitis was too small that may produce bias in subgroup analysis, we just put the table below the response letter instead of manuscript.

Comment 3:

How about severity of IBD?

Answer: 

The severity of IBD was defined by the clinical signs such as frequency of diarrhea, abdominal pain, fever, anemia…etc, which were not included in our database. We had modified at Discussion section, line 273~275 on page 12.

Comment 4:

Please add data of PPI/H2RA in Table.

Answer: 

We had added data of PPI/H2RA in Table 1 and S1 table. The use of PPI significantly increased in PUD+HPRx in general population and DM groups due to H. pylori treatment. The use of H2RA significantly increased in PUD+HPRx in general population and DM groups, which may owing to more upper GI symptoms in patients with H. pylori infection.

---

## [Decision Letter · Decision Letter 2]

1 Mar 2022

Effects of Helicobacter pylori treatment on the incidences of autoimmune diseases and inflammatory bowel disease in patients with diabetes mellitus

PONE-D-21-21394R2

Dear Dr. Kun-Der Lin,

We’re pleased to inform you that your manuscript has been judged scientifically suitable for publication and will be formally accepted for publication once it meets all outstanding technical requirements.

Kind regards,

Yoshio Yamaoka

Academic Editor

PLOS ONE

Additional Editor Comments (optional):

This revised version is well-written.

Reviewers' comments:

Reviewer's Responses to Questions

**Comments to the Author**

1. If the authors have adequately addressed your comments raised in a previous round of review and you feel that this manuscript is now acceptable for publication, you may indicate that here to bypass the “Comments to the Author” section, enter your conflict of interest statement in the “Confidential to Editor” section, and submit your "Accept" recommendation.

Reviewer #2: All comments have been addressed

2. Is the manuscript technically sound, and do the data support the conclusions?

Reviewer #2: Yes

3. Has the statistical analysis been performed appropriately and rigorously? 

Reviewer #2: Yes

4. Have the authors made all data underlying the findings in their manuscript fully available?

Reviewer #2: Yes

5. Is the manuscript presented in an intelligible fashion and written in standard English?

Reviewer #2: Yes

6. Review Comments to the Author

Reviewer #2: This version of this study was well revised according to Reviewer’s suggestions and recommendations.

7. PLOS authors have the option to publish the peer review history of their article (what does this mean?). If published, this will include your full peer review and any attached files.

Reviewer #2: No

---

## [Editor Report · Acceptance letter]

12 May 2022

PONE-D-21-21394R2 

Effects of *Helicobacter pylori* treatment on the incidences of autoimmune diseases and inflammatory bowel disease in patients with diabetes mellitus 

Dear Dr. Lin:

I'm pleased to inform you that your manuscript has been deemed suitable for publication in PLOS ONE. Congratulations! Your manuscript is now with our production department. 

Kind regards, 

on behalf of

Professor Yoshio Yamaoka 

Academic Editor

PLOS ONE